# Effects of Citrulline Supplementation on Different Aerobic Exercise Performance Outcomes: A Systematic Review and Meta-Analysis

**DOI:** 10.3390/nu14173479

**Published:** 2022-08-24

**Authors:** Aitor Viribay, Julen Fernández-Landa, Arkaitz Castañeda-Babarro, Pilar S. Collado, Diego Fernández-Lázaro, Juan Mielgo-Ayuso

**Affiliations:** 1Glut4Science, Physiology, Nutrition and Sport, 01004 Vitoria-Gasteiz, Spain; 2Institute of Biomedicine (IBIOMED), University of Leon, 24071 Leon, Spain; 3Physical Education and Sports Department, Faculty of Education and Sport, University of the Basque Country (UPV/EHU), 01007 Vitoria-Gasteiz, Spain; 4Health, Physical Activity and Sports Science Laboratory, Department of Physical Activity and Sports, Faculty of Education and Sport, University of Deusto, 48007 Bilbao, Spain; 5Department of Cellular Biology, Genetic, Histology and Pharmacology, Faculty of Health Sciences, University of Valladolid, Campus of Soria, 42003 Soria, Spain; 6Neurobiology Research Group, Faculty of Medicine, University of Valladolid, 47005 Valladolid, Spain; 7Department of Health Sciences, Faculty of Health Sciences, University of Burgos, 09001 Burgos, Spain

**Keywords:** Citrulline, ergogenic aids, physical performance, nitric oxide, aerobic, endurance

## Abstract

Supplementation with Citrulline (Cit) has been shown to have a positive impact on aerobic exercise performance and related outcomes such as lactate, oxygen uptake (VO_2_) kinetics, and the rate of perceived exertion (RPE), probably due to its relationship to endogenous nitric oxide production. However, current research has shown this to be controversial. The main objective of this systematic review and meta-analysis was to analyze and assess the effects of Cit supplementation on aerobic exercise performance and related outcomes, as well as to show the most suitable doses and timing of ingestion. A structured literature search was carried out by the PRISMA^®^ (Preferred Reporting Items for Systematic Reviews and Meta-Analyses) and PICOS guidelines in the following databases: Pubmed/Medline, Scopus, and Web of Science (WOS). A total of 10 studies were included in the analysis, all of which exclusively compared the effects of Cit supplementation with those of a placebo group on aerobic performance, lactate, VO_2_, and the RPE. Those articles that used other supplements and measured other outcomes were excluded. The meta-analysis was carried out using Hedges’ g random effects model and pooled standardized mean differences (SMD). The results showed no positive effects of Cit supplementation on aerobic performance (pooled SMD = 0.15; 95% CI (−0.02 to 0.32); *I*^2^, 0%; *p* = 0.08), the RPE (pooled SMD = −0.03; 95% CI (−0.43 to 0.38); *I*^2^, 49%; *p* = 0.9), VO_2_ kinetics (pooled SMD = 0.01; 95% CI (−0.16 to 0.17); *I*^2^, 0%; *p* = 0.94), and lactate (pooled SMD = 0.25; 95% CI (−0.10 to 0.59); *I*^2^, 0%; *p* = 0.16). In conclusion, Cit supplementation did not prove to have any benefits for aerobic exercise performance and related outcomes. Where chronic protocols seemed to show a positive tendency, more studies in the field are needed to better understand the effects.

## 1. Introduction

Aerobic or cardiorespiratory endurance exercise places high physiological and metabolic demands on the human body’s organic regulatory systems [1]. From a physiological point of view, performance in cardiorespiratory endurance sports, where the effort generally lasts longer than 5 min, depends primarily on three related factors: maximal oxygen uptake (VO_2_ max), which can be defined as the maximum oxygen-carrying capacity of a subject; the individual lactate threshold (ILT), which has many names and is related to different physiological concepts but which, in short, refers to the intensity of exercise at which the concentration of lactate in the blood increases exponentially compared to basal levels due to the inability to be oxidized in the mitochondria; and the efficiency or running economy, defined as the oxygen consumption or energy required by an individual to maintain a given running pace [2,3,4,5,6]. In addition, the importance of psychological sensation is of the utmost importance in sport, as it is considered a form of biofeedback integrated into the information and effort regulation system between the central command and the local peripheral mechanisms [7,8].

In competitive sport, a 1–2% improvement in performance can make the difference between winning and losing a competition and is, therefore, a critical difference [9]. To improve sports performance, athletes work with complex, carefully planned training and nutrition systems [10,11]. Nutrition and supplementation are therefore important tools in optimizing performance, especially in endurance and aerobic disciplines [6,12]. In their search for the ultimate performance improvement, athletes often make use of a variety of sports supplements [13]. Among the most-used supplements, proteins and amino acids are the most consumed, with a frequency of use of 35–40% [14]. However, the use of nutritional supplements with functions related to Nitric Oxide (NO) synthase has increased considerably in the field of sport, due to evidence linking their intake to positive effects on sports performance [12,15,16].

The ergogenic interest in NO-related supplements and their different ways of obtaining them focuses on the potential effects that this bioactive compound can have on the body. These include health-enhancing effects such as the regulation of blood flow and blood pressure, the maintenance of gastric integrity, and protection against ischemic tissue damage [17,18,19]. They have also been found to be related to improved performance in different types of disciplines and through different supplements, improving vasodilation and angiogenesis, causing an increase in mitochondrial respiration and biogenesis, positively impacting glucose uptake, optimizing oxygen regulation, and improving muscle contraction [20,21,22,23,24,25,26].

Among the supplements linked to these effects are two non-essential amino acids, namely Arginine (Arg) and Citrulline (Cit) [16,27]. Cit is an amino acid abundantly found in watermelon (*Citrullus vulgaris*) and can be synthesized from Arg and another amino acid called Ornithine (Orn), with plasma glutamine and Arg itself being the main precursors [27,28]. Likewise, Cit is a precursor to Arg [29,30]. Therefore, Cit plays an essential role in the bioavailability of Arg and also in the subsequent synthesis of NO [29]. However, unlike Arg, Cit passes directly into the kidneys without being catabolized by the enzyme arginase [31]. Cit itself can inhibit arginase action and limit the catabolism of Arg to Orn [32]. It is therefore hypothesized that dietary Cit intake might be more efficient in increasing Arg bioavailability and NO synthesis than Arg supplementation itself [28,32].

For this reason, Arg and Cit supplementation has been used in athletes from different disciplines and conditions to improve athletic performance [20,33,34,35,36,37]. A recent systematic review and meta-analysis [24] showed a positive effect of Arg supplementation in different sports disciplines, depending on the energy metabolism used. However, evidence for the effect of Cit on aerobic and cardiorespiratory endurance exercise is inconsistent to date. There are insufficient studies that have qualitatively and quantitatively analyzed its effect on physical performance and the different variables involved.

The physiological demands of endurance sports disciplines are directly related to the ability of the cardiovascular system to provide oxygen and nutrients to the muscles, and the functionality of the peripheral (muscular) structures to obtain the required energy, that is, adenosine triphosphate (ATP), as efficiently as possible. Therefore, the training, nutrition, and supplementation methods used are of particular importance for athletes in these disciplines. They are therefore of great interest considering the potential beneficial effects that the intake of supplements that exogenously and endogenously stimulate NO synthesis have in relation to the demands required by long-duration exercise [20]. The increased synthesis of this bioactive compound could ultimately result in multiple beneficial effects [17,20,38,39,40].

Despite the promising evidence of Cit supplementation being used in both high intensity, power, and strength exercise [41,42], as well as in long-term exercise [43,44,45], there are no conclusive results about its possible effects. Although acute supplementation with Cit (single or multiple doses within a single day) has been shown to significantly increase plasma levels of Arg and NO, no improvement in athletic performance has been observed [46,47]. Similarly, no effect was shown on VO_2_ max, anaerobic threshold, time to exhaustion, and the performance parameters analyzed after doses of Cit of between 6 and 12 g administered 1–2 h before exercise [43,44,48]. In contrast, supplementation using chronic protocols has shown more promising results in relation to endurance performance variables. For example, 7-day Cit supplementation of 2.4 g/day and 6 g/day of Cit improved performance in a 4 km time trial [49] and VO_2_ kinetics and fatigue tolerance in strenuous exercise on a cycle-ergometer [45], respectively. However, one aspect to consider in the interpretation of these results is the level of the participants in the studies. The best results regarding supplementation with ergogenic aids following the NO route have been mainly related to untrained subjects or to participants who are at low competitive levels, whereas no major differences have been found in athletes with high training levels [20].

With this in mind, this systematic review and meta-analysis aim to qualitatively and quantitatively analyze the impact of Cit supplementation on aerobic performance and related variables such as lactate, VO_2_ kinetics, and the RPE to analyze the current evidence in an orderly and systematic way, to extend the current understanding of the efficacy, effects, and mechanisms behind sports supplementation.

## 2. Methods

### 2.1. Literature Search Strategies

This systematic review and meta-analysis were conducted by the PRISMA (Preferred Reporting Items for Systematic Reviews and Meta-Analyses) guidelines [50] and the PICOS inclusion criteria definition model: “P” (sample or population): “active”; I (intervention): “impact of Citrulline on sports performance”; C (comparison): “Citrulline supplementation compared to Placebo”; O (outcomes): “aerobic sports performance and related variables”; S (study design): “double-blind, crossover, randomized, clinical trial” [51]. A review of the scientific literature was carried out, including those studies that analyzed the effect of Cit supplementation on aerobic sports performance and its related variables during exercise, namely, lactate, the rate of perceived exertion (RPE), and oxygen uptake kinetics (VO_2_). The search was conducted using the respective search engines of the different scientific databases, without applying any temporal filter, thus covering all entries from the first studies on the subject in 2006 until the latest published to date in 2021. Specifically, the PubMed/MEDLINE, Web of Science (WOS), and Scopus databases were used, using the following Boolean search equation: L-Citrulline [All Fields] OR “Citrulline” [All Fields] OR “watermelon juice” [All Fields] AND supplementation [All Fields] AND ((“sports” [MeSH Terms] OR “sports” [All Fields] OR “sport” [All Fields] OR “sport” [All Fields]) OR (“exercise” [MeSH Terms] OR “exercise” [All Fields])) AND (“endurance” [All Fields] OR “performance” [All Fields] OR “aerobic” [All Fields] OR “lactate” [All Fields] OR “perceived exertion” [All Fields] OR “oxygen kinetics” [All Fields]).

Using this equation, relevant articles in the field were selected by applying a “snowball” search strategy through the references obtained in the initial search. All titles and abstracts were cross-checked and analyzed to identify duplicates or any potential errors, as shown in the flow chart. To do this, the “duplicate search” option in Mendeley Reference Manager was used. All titles and abstracts were also screened to consider whether reading the full articles would be necessary. The search, screening, and selection of published studies were carried out independently by two authors (A.V.M. and J.M.-A.), according to the proposed inclusion and exclusion criteria.

### 2.2. Inclusion and Exclusion Criteria

The inclusion criteria applied in this systematic review and meta-analysis for the selection of the articles to be considered in the study were as follows:-Studies performed on humans.-Studies with an appropriate design, using a double-blind, randomized, crossover methodology.-Studies with 2 well-differentiated groups comparing the effects of Citrulline supplementation (experimental group) versus a placebo (control group).-Studies were conducted on active people, recreational athletes, well-trained athletes, and elite or healthy professional athletes, both men and women.-Studies in which the comparison of placebo versus supplementation was only with Citrulline and other possible masking compounds, avoiding the intake of different ergonutritional aids at the same time.-Studies specifying the amount and type of supplementation used.-Studies measuring aerobic physical performance using continuous tests lasting 5 min or more, following the criteria relating to the measurement of VO_2_ max.-Studies measuring blood lactate, the RPE, and different variables of VO_2_ kinetics.-Studies published in any language.-Studies for which the full manuscript was available.

The following exclusion criteria were used to discard articles found in the initial search:-Studies in which supplementation did not involve Citrulline alone, including those using a combination of different nitric oxide pathway supplements.-Studies in which there was no real presence of a control group ingesting a substance considered to be a placebo.-Studies in which other supplements or ergonutritional aids that might interfere with performance were used in the interventions, considering the scientific evidence in the literature, such as Caffeine, Beta-Alanine, Creatine, etc.-Studies in which performance was measured using tests lasting less than 5 min and/or measuring variables related to anaerobic performance.-Studies conducted on an unhealthy, sick, or injured population.

Finally, no filters were applied concerning the gender, level, or sporting discipline of the subjects participating in the studies, and any research that analyzed the proposed variables in any situation that met the inclusion criteria was considered to be valid.

### 2.3. Study Selection

After searching, the various studies were screened and unilaterally selected for analysis by the author. After applying the inclusion and exclusion criteria, data were extracted from each study as shown in the “Results” section. The source of the studies, including authors, year of publication, and citation information, the sample size and characteristics of the participants (level, age, and gender), the administration of the supplement to be analyzed and the placebo to be controlled for (dose, timing), the final variables to be reviewed (test, intensity, and characteristics), and, finally, the main conclusions of each study were extracted by the author with the help of a spreadsheet (Microsoft Inc., Seattle, WA, USA). Subsequently, the data presented were analyzed and several studies were excluded for relevant reasons until the studies to be included in this systematic review and meta-analysis were obtained.

### 2.4. Measuring Variables

The available scientific literature on Cit supplementation and its effect on aerobic physical exercise was examined in terms of performance in the physical tests conducted, the analysis of lactate as a determining energy metabolite, the RPE, and VO_2_ kinetics as a physiological factor limiting aerobic performance. To quantify aerobic sports performance, measurement variables classified according to the duration of the tests were used. The 5 min test appeared to be a reliable test for determining maximal aerobic speed, related to VO_2_ max, and therefore suggested the time that a subject can maintain the lowest intensity related to this physiological requirement [52,53]. Taking into account that VO_2_ max marks a subject’s maximum aerobic power, the author established tests lasting ≤ 5 min as time criteria for analyzing aerobic sports performance in this systematic review and meta-analysis.

More specifically, several tests were included in the performance analysis to determine aerobic performance, both on the cyclo-ergometer and treadmill, according to the proposed criteria: incremental and sustained tests to exhaustion; Bruce protocol; treadmill test at sub-maximal intensity (40% of HHR); constant work test at 70% of VO_2_ max; moderate intensity test (90% of HHR); constant work test at 70% of VO_2_ max; moderate intensity test (90% of VO_2_ max); moderate intensity test (90% of GET) followed by a high-intensity test on the day (GET + difference between GET and VO_2_ max work); 6 min constant work test (70% VO_2_ max) followed by a 30 s all-out test at 70% of VO_2_ max; a 30 s all-out test; 10 repetitions of 15 s at maximum intensity (sprint); 5 min rest and an incremental test to exhaustion at 100% of individual peak power; a half-marathon race; a 4 km and 40 km individual time trial; and an SRT test comprising a 6 × 1-min sprint at 120% of maximum power.

Likewise, for the analysis of lactate, the measurement of this biomarker was analyzed in the different studies in the same way as for the RPE, using the Modified Borg Scale (0–10) or Borg Rating of Perceived Exertion Scale (6–20). In turn, different related variables were used to analyze the responses for VO_2_ kinetics. These included: net oxygen expenditure, mean response time (MRT), oxygen deficit, mean pulmonary VO_2_ response time, mean overall pulmonary gas response time, tissue oxygenation index, muscle oxygenation, pulmonary VO_2_, VO_2_ max, gas exchange defined threshold, and peak VO_2_.

For the statistical analysis of the quantitative synthesis (meta-analysis), the sample size, means, and standard deviations of the different variables were extracted for the two groups compared: the Cit supplementation group and the placebo group, respectively. In those cases where data were not found in a numerical form, the results were extracted by analyzing the figures and graphs using the Review Manager (Revman) v5.4 (Copenhagen: The Nordic Cochrane Centre, The Cochrane Collaboration, Copenhagen, Denmark), which estimates the results based on the process of coding the pixels of the respective images.

### 2.5. Publication Bias

Publication biases were assessed using Egger’s statistical test, whereby publication bias occurs when *p* < 0.05 [54]. To visualize the calculation and interpret the results, the following funnel plots were performed, followed by Egger’s statistic to confirm or refute publication bias of each article included in this systematic review (Figure 1). Egger’s analysis and the funnel plots suggest low publication bias in the 4 categories analyzed.

### 2.6. Assessing the Quality of Investigations

This systematic review was prepared by the Cochrane Collaboration Guidelines [55]. The quality assessment of the different articles was carried out independently by two authors (A.V.M. and J.M.-A.). The risk of bias analysis was separated into six distinct domains, as shown in Figure 2: selection biases (random sequence generation, allocation concealment); biased elaboration (blinding of participants and researchers); detection bias (blinding of outcome assessment); attrition bias (incomplete outcome data); reporting bias (selective reporting); and other types of biases. Regarding the assessment of the domains, it was considered to be: “low” if the criterion had a low risk of bias and it was unlikely to alter the results, or “high” if the criterion had a high risk of bias and it was likely to seriously undermine the results. If the risks of bias were not known, they were categorized by labeling them as “unknown”, which signifies doubts as to the results.

Figure 3, below, is a clear visual representation of the analysis bias of each article included in this systematic review and meta-analysis.

## 3. Results

### 3.1. Main Search

The initial literature search using the electronic databases showed a total of 94 records matching the Boolean search engines proposed in the search equation. Additionally, eight records were found using other reference lists, yielding an initial total number of 102 articles. However, only 10 met all of the inclusion and exclusion criteria and were therefore included in the systematic review and meta-analysis. Following the process of duplicate analysis, 17 articles were excluded, leaving 85 references for screening. Of these, 28 were excluded for different reasons: 21 references used a non-human sample and seven were either narrative or systematic reviews and/or meta-analyses. Therefore, 57 records were screened for full-text analysis and assessed for eligibility. After the assessment, a total of 48 records were excluded for various reasons related to the inclusion and exclusion criteria. Of these, 19 studies were rejected because they had used more than one supplement in conjunction with the ILC, nine studies were excluded because they employed other unrelated supplements, six were eliminated because they used subjects unrelated to the inclusion criteria, and 14 were excluded because they analyzed variables unrelated to those proposed in the methodology of this paper. Following this suitability analysis, 10 articles were chosen for qualitative synthesis (systematic review). Subsequently, after analyzing and extracting the numerical and/or image data, these 10 were included in the final quantitative synthesis (meta-analysis). Therefore, the 10 articles that met all the proposed inclusion and exclusion criteria were included in both the systematic review and the meta-analysis [43,44,45,48,49,56,57,58,59,60].

The flow chart in Figure 4, below, shows the sequence of actions that were carried out during the main search, following the PRISMA guidelines [50] and justifying the reasons for the exclusion of the articles removed from the synthesis.

### 3.2. Study Characteristics

Table 1 shows the study characteristics (design, gender of the subjects, as well as the type, dose, and time of supplementation) of the studies included in the systematic review and meta-analysis. The total sample analyzed comprised 10 articles totaling 173 subjects. All 10 articles used a double-blind, randomized, placebo-controlled, crossover methodology [43,44,45,48,49,56,57,58,59,60]. None of the studies presented any conflict of interest concerning their intervention and the supplementation used. Of the 10 studies analyzed, four of them involved recreational athletes [45,57,58,59], three were well-trained and/or elite sportspeople [44,49,60], two studies involved active adults [43,56], and the remaining article was concerned with intercollegiate athletes [48]. In the 10 articles [43,44,45,48,49,56,57,58,59,60], two comparative groups (Cit and placebo) were separated, while in two of them, an additional group was used to compare different dosages [43,48]. A group was used in one of them to compare the timing of Cit intake and the time of ingestion [43].

A total of five studies used an acute dose of supplementation [43,44,48,58,59], of which two used Cit 1 h before the test [44,61], one study 1 h to 2 h before the experimental test [48], another study 2 h before, [59] and the remaining study 3 h before the test [43]. In contrast, six studies used a chronic supplementation methodology ranging from 1 day to 16 days [43,45,56,57,59,62]. Of these, Hickner et al. [43] used 1-day supplementation, while Bailey et al. [45] used 6 days of supplementation. Three studies [49,56,60] supplemented the Cit for 7 days, and the remaining [57] for a longer period of 16 days.

In all the studies reviewed, the dose of Cit supplementation was absolute, with no studies using doses relative to body mass or other variables, as is the case with other supplements [63]. L-Citrulline was used as a supplement in six studies [43,45,48,49,56,60], while in one [63], participants ingested watermelon juice. In two other studies [57,59], a mixture of both supplements (watermelon juice enriched with L-Citrulline) was used, whereas the two remaining studies [44,58] involved an intake of Citrulline Malate. Regarding supplementation doses, four studies [45,48,56,60] used 6 g, depending on their timing; two studies [57,59] used 3.4 and 3.45 g; one study [48] used 1 g; another [49] 2.4 g; one article [43] used 3 g; one article used 8 g [58]; and the remaining two entailed intakes of 9 g [43] and 12 g of a Cit supplement [44].

### 3.3. Effect of Citrulline on Aerobic Performance (≤VO_2_ Max)

Table 2 shows the results of the different tests conducted in the selected studies that determined sporting performance at an intensity less than or equal to VO_2_ max (≤VO_2_ max) and the variables measured.

Two studies used an incremental test to exhaustion in both analysis groups [43,48]. One of them used the Bruce protocol [48]. Cunniffe et al. [44] employed a test based on 15 pre-sprint sprints, followed by a 5 min rest and then a test at 100% of peak power to exhaustion. Along the same lines, but at an intensity of 90% of VO_2_ peak, Gills et al. [58] used an exhaustion test until a cadence was reached that was below 40 rpm. Bailey et al. [45] used two different tests in their 2015 article, the first being of moderate intensity (90% of Gas Exchange Threshold) (GET) and the second of high intensity (GET + difference between work in GET and VO_2_ max), whereas in the 2016 study [57], two different protocols were used for each group: one based on constant work on a cycle-ergometer at 70% of VO_2_ max to exhaustion, and the other consisting of 6 min of constant work (70% VO_2_ max) + 30 s all-out on a cycle-ergometer. However, one study [60] used a 40 km individual time trial as a test followed by six sprints of 1 min at 120% of maximum power. The remaining article [59] measured the results of supplementation in a half-marathon race.

Of the articles analyzed, Bailey et al. [45] found improvements in work tolerance test time, as well as in total work test completed. In the same vein, Hickner et al. [43] showed a benefit in time to exhaustion in the two groups analyzed and a benefit in time to exhaustion in the two groups analyzed. Stanelle et al. [60] found significant improvements in the time to complete the 40 km time trial. In addition, one study showed improvements in 10” total work after a series of constant exercises at 70% of VO_2_ max [57]. In contrast, the rest of the studies [44,48,58,60] found no significant improvements in the variables analyzed, concluding that Cit supplementation did not induce positive effects in the tests performed.

Regarding the meta-analysis, Figure 5 shows the results of the meta-analysis that assessed the impact of Cit supplementation on aerobic performance. It can be seen that CWB had no significant effect on this sports performance variable (*p* = 0.12). The meta-analysis reported low inconsistency between studies (I = 0%; *p* = 1.00). Two variables showed a large or moderate effect, these being cadence during repeated sprint work [60] and tolerance to exercise time at high intensity on a cycle-ergometer [45], respectively. Four variables showed a small effect when comparing supplementation with placebo and the remaining 19 variables showed a negligible effect. Therefore, the overall meta-analysis of the effect of Cit on aerobic performance showed a negligible effect on that performance variable (pooled SMD = 0.12; 95% CI (−0.03 to 0.27)).

### 3.4. Effect of Citrulline on the RPE

The effect of Cit supplementation on the rate of perceived exertion (RPE) of the analyzed items is shown in Table 3. The study by Hickner et al. [43] measured the submaximal RPE using the incremental protocol to exhaustion on the treadmill. The results showed an increase in the RPE that was related to the improvement in time to exhaustion reported in Table 3. Martinez et al. [59] measured the RPE at the end of a half-marathon race, with no significant differences between the placebo group and the experimental group that ingested Cit as a supplement. Along the same lines, Suzuki et al. [49] found no statistically significant differences in the RPE after a 4 km individual time trial. Stanelle et al. [60] measured the RPE in the two tests used in their experiment, both in the 40 km individual time trial and in the repeated sprint work that followed. The results showed a statistically significant and superior effect in the group supplemented with CWB compared to the group that ingested a placebo, results that were consistent with both the positive effect found in the time trial performance and the absence of reported performance differences in the subsequent sprint exercise.

According to the quantitative analysis, Cit supplementation had non-significant effects on the RPE in the studies analyzed (*p* = 0.90), as shown in Figure 6. The meta-analysis also describes low inconsistency levels among the selected studies (I = 49%; *p* = 0.07). The study conducted by Suzuki et al. [49] showed a strong effect on the perception of muscle fatigue and a moderate effect on pedaling ease. However, the remaining studies analyzed had a negligible impact on the RPE. Overall, the meta-analysis found a negligible effect on the RPE when comparing Cit supplementation and placebo (pooled SMD = −0.03; 95% CI (−0.43 to 0.38)).

### 3.5. Effect of Citrulline on VO_2_ Kinetics

Table 4 shows the studies included in the systematic review and meta-analysis that analyzed the effects of Cit supplementation on VO_2_ kinetics using different endpoints.

Of the six studies that looked at the impact of Cit on VO_2_ response, two [45,56] analyzed pulmonary O_2_ uptake (VO_2_) mean response time (MRT) and found statistically significant effects that showed an improvement in this variable. However, while they found statistically significant effects that showed an improvement in this variable, Ashley et al. [56] only found these results in one of the groups, with no benefits having been reported in the group of people over 60 years of age. Additionally, two studies looked at muscle oxygenation [57] and tissue [45] in submaximal and maximal exercises and found significant improvements related to these variables. In contrast, Cutrufello et al. [48] did not find significant results in VO_2_ max. or anaerobic threshold, nor did Bailey et al. [57] in pulmonary VO_2_, or Hickner et al. [43] in VO_2_ peak. Likewise, Ashley et al. [56] reported no significant differences in net oxygen expenditure, nor did the remaining study [49] in VO_2_ kinetics.

Figure 7 shows the results of the meta-analysis about VO_2_ kinetics after Cit supplementation, where no significant effect was found for this variable (*p* = 0.94). A meta-analysis of the selected studies found low inconsistency levels (I = 0%; *p* = 0.91). One study showed a large effect on a single measured variable, namely, the pulmonary gas exchange mean response time after the high-intensity cycle-ergometer exercise [45]. In addition, two variables showed a small effect [56], while the remaining 17 were related to a negligible effect. Therefore, the meta-analysis showed a negligible effect on VO_2_ kinetics by Cit supplementation (pooled SMD = 0.01; 95% CI (−0.16 to 0.17).

### 3.6. Effect of Citrulline on Lactate

Four of the nine studies included in the systematic review and meta-analysis looked at lactate as the main metabolite of glycolysis [64] after the ingestion of Cit and placebo. Table 5 shows these studies, including their methodology and their main conclusions regarding lactate measurement.

In three studies [44,57,59], an increase in blood lactate concentration was shown in the experimental group compared to the control group. Martínez et al. [59] analyzed lactate after a submaximal effort of 40 km, whereas Bailey et al. [57] did so after sub-maximal and maximal effort, and Cunniffe et al. [44] after a sprint effort. The remaining study, carried out by Hickner et al. [43], did not show a significant increase in lactate in either of the two experimental groups in which different doses and timings of Cit supplementation were used.

The meta-analysis did not show any significant positive effect on lactate compared to placebo (*p* = 0.16). The studies analyzed found low inconsistency between them, as shown by the meta-analysis (I = 0%; *p* = 0.94). Figure 8 shows the results obtained. It indicates the moderate effect of lactate found in the study by Martínez et al. [59] after participants completed a half-marathon race. However, two variables from one study [57] showed a small effect, and the remaining three, obtained from two studies, showed a small effect [43,44] and a negligible effect. Therefore, the meta-analysis showed a small effect of Cit supplementation on lactate (SMD grouped = 0.25; 95% CI (−0.10 to 0.59)).

## 4. Discussion

The main objective of this systematic review and meta-analysis was to analyze and summarize the current evidence on the effects of Cit supplementation on aerobic sports performance and related variables such as lactate, VO_2_ kinetics, and the RPE. Overall, the results obtained indicate that Cit had no beneficial effects on aerobic performance, lactate concentration, VO_2_ kinetics, and/or the RPE in the tests performed. The doses and timing of intake differed between the studies analyzed, suggesting that there may be different mechanisms at play depending on these variables. Regarding timing, chronic supplementation for 6–16 days seemed to have positive, albeit statistically non-significant effects, while acute supplementation (1 h–3 h before the test) did not have a positive impact. In this sense, both a chronic dose of Cit between 3.4 and 6 g/day for 6–16 days and an acute dose (1 h–3 h before) of between 1 and 12 g did not significantly improve aerobic performance. Similarly, Cit did not improve the RPE score under the same supplementation conditions, and VO_2_ kinetics were not significantly altered. Supplementation with 3–12 g of acutely and chronically ingested Cit showed a clear trend of increased lactate, although the overall effect was only small (*p* = 0.16).

Considering the already known role of Cit in the synthesis and production of NO and the possible effects associated with this compound, the potential impact of its supplementation on physical performance and associated variables in different types of exercise and using different tests has been studied in recent years [59,65]. A positive impact on various training adaptations related to other NO pathway supplements has also been suggested [24,26]. However, studies examining such effects are scarce. As a result, the evidence for Cit supplementation is controversial and, therefore, it is not considered to be an ergogenic aid backed by strong scientific evidence [12,15]. Therefore, although the most commonly used dose was between 3.4 and 6 g, there was no consensus on doses and protocols for use to improve sports performance. The results obtained in this systematic review and meta-analysis are similarly controversial, as they showed no uniform impact on those variables measured after the ingestion of this amount. Likewise, greater intakes (9–12 g) that have shown positive effects for other related supplements such as Arg [24] have not shown such benefits in this study. Similarly, doses of less than 3.4 g have not been found to have a positive impact on performance or associated variables. However, it is worth highlighting the possible differences associated with the intake timing, as a slight additional benefit seems to be shown in those protocols based on chronic supplementation compared to acute supplementation. This could be explained by the results already obtained in studies with other supplements of the NO pathway, such as beetroot juice or nitrate, where chronic supplementation seems to be necessary to generate adaptations at a higher level related to NO [25]. About acute supplementation, it is well known in the literature that nitrate and nitrite levels peaked after 2.5–3 h [39] and Arg around 30–90 min after ingestion [66]. Taking this into account, and considering the role played by Cit in the synthesis of Arg together with NO from the nitrate–nitrite–NO pathway, acute supplementation should propose protocols by these proven effects. The present study suggests different effects between one protocol and the other; therefore, there is a need to study both types of supplementation to understand which methodology could be more appropriate for the optimization of sports performance.

The long-term performance or endurance depends on the ability to efficiently synthesize ATP primarily through oxidative pathways, as well as to produce and oxidize lactate through oxidative phosphorylation [1,67,68]. It is primarily influenced by different variables such as lactate concentration, which is an indirect measure of cellular metabolism; VO_2_ kinetics, which determines physiological demands; and the RPE, as a subjective perception of the effort made. The use of different nutritional and supplementation methods can therefore make a significant difference in sporting performance [6,12]. Moreover, taking into account the physiological effects related to NO at the ventilatory level, as well as energy efficiency and muscle contraction, using supplements such as Cit to enhance its endogenous synthesis could be an interesting strategy [25]. While there are studies that have shown significant improvements in their participants [43,45,57,60], there are several other papers whose results have not shown any positive effect [48,49,59]. Consequently, Cit supplementation, in isolation, is not currently recommended for aerobic performance enhancement.

### 4.1. Effect on Aerobic Sports Performance

Long-term sporting performance depends primarily on high aerobic capacity, which allows the cardiovascular system to supply sufficient oxygen to meet the needs of exercise, and high metabolic flexibility that facilitates the use of energy substrates according to the demands of the moment [1,69]. For example, the use of a nutritional strategy based on higher or lower carbohydrate intake can make a significant difference in winning or losing a competition such as an asphalt marathon [6]. The use of an effective ergo-nutritional aid can also make an important difference [6]. In this regard, Cit supplementation has been shown to have beneficial effects associated with aerobic capacity. Bailey et al. [45] found a significant improvement in exercise tolerance time and work completed in two cycle-ergometer tests at different intensities (moderate and severe) after chronic supplementation for 6 days, with the last dose taken 90 min before exercise (6 g of Cit). Similarly, Stanelle et al. [60] reported clear benefits in the time to complete a 40 km individual time trial, as well as in average power output, following a similar supplementation protocol but with a 1-day longer duration. However, the same study showed no positive effects in a series of six sprints performed immediately after completing the time trial. In another similar study, in which the timing of the dose was longer (16 days) but smaller (3.4 g/day of Cit), no improvement in time to exhaustion was found in a constant work test at 70% of VO_2_ max [57]. These results could be explained by the lower dose ingested in the latter study, which was almost half of the dose with positive effects.

Using an acute supplementation methodology, Hickner et al. [43] found a significant improvement in time to exhaustion in two groups with different doses of Cit: 9 g/day ingested over 24 h with a final intake 3 h before the test, and a single dose of 3 g/day taken 3 h before the test. Martínez et al. [59] found no positive results in the final time of a half-marathon after the use of a similar dose of 3.45 g (in the form of watermelon) was ingested 2 h before exercise, and a single dose of 3 g/day was taken 3 h before the test. The same results were obtained in another study that compared two different doses (6 g and 1 g) ingested 1–2 h before an incremental protocol to exhaustion among intercollegiate players, without showing any improvement associated with time to exhaustion [48]. While low doses could be the reason for these results, a study conducted by Cunniffe et al. [44] showed that even a dose of 12 g ingested 60 min before an intermittent test followed by an exhaustion test did not provide any benefits in time to exhaustion, average power, work done, distance covered, or average speed. It seems that acute supplementation with Cit does not yield any improvement in aerobic sports performance, regardless of the dose used and the level of the sample. In the same vein, it is particularly interesting to consider the pharmacokinetics of Cit to study the best timing methodology for optimization in the same way as for other supplements such as Arg.

Acute and chronic supplementation may have different physiological mechanisms of action and may also be related to the doses of Cit supplemented. The results of this systematic review and meta-analysis show no clear beneficial effects for either strategy (*p* = 0.08), although chronic supplementation with an amount close to and/or above 6 g/day may show a trend towards improvement, in contrast to the acute approach.

### 4.2. Effect on the RPE

The rate of perceived exertion is a powerful indicator of the internal load that a subject endures when exercising. It is also related to multiple other variables that vary along with the duration and intensity of exertion, such as different physiological variables related to oxygen consumption and heart rate, metabolic variables such as lactate concentration or glycogen depletion, and other specific phenomena such as muscle damage, temperature, or dehydration [70,71]. Its quantification by means of a subjective rating scale such as the Borg scale [72] has been shown to be valid for detecting physiological and metabolic changes during exercise, as well as for monitoring internal load during training [73,74]. The sensation of exertion of an exerciser can be modified by different tools related to nutrition and supplementation, such as carbohydrate intake, fluid and electrolyte intake, or supplementation with substances such as caffeine or creatine [75]. In the same vein, NO-related supplements such as nitrate or beetroot juice have already been shown to have direct and beneficial effects on the RPE during and after exercise [61,76].

Regarding Cit supplementation, Hickner et al. [43] showed an increase in the submaximal RPE of two groups of ten and seven physically active men and women during an incremental test to exhaustion after ingesting 9 g/day and 3 g/day, respectively. The same results were reported by another study using an intermediate dose of 6 g/day in nine well-trained male triathletes and competitors competing in a 40 km individual time trial [60]. These results in the RPE correspond with an improvement in aerobic capacity in both studies, demonstrating that, along with an improvement in sports performance, there could be a positive effect of Cit on the RPE. On the contrary, two other studies did not find significant differences between their results. Martínez et al. recorded no change in the RPE of 22 male amateur runners after the acute supplementation of 3.4 g of Cit in a half-marathon race. With similar results, Suzuki et al. [49] found no improvement in the RPE of 22 well-trained men who chronically ingested 2.4 g of Cit for 7 days prior to competing in a 4 km individual time trial.

While a lower score on the RPE scale may lead to an improvement in sports performance due to a lower perception of fatigue, an increase in the RPE scale may at the same time lead to an increase in exercise intensity and thus to higher self-demand in exercise, which could lead to better performance. However, such an increase should be related to an objective improvement in performance measured by other variables, as was the case in the studies analyzed in this systematic review and meta-analysis [43,60]. These results could be explained by a multitude of factors related to the effects of Cit and NO on performance, including increased muscle oxygenation, reduced oxygen consumption, and improved muscle contraction [20]. Despite the positive results, it cannot be concluded that Cit supplementation improves the RPE during endurance exercise due to the limited literature available and the results obtained in this meta-analysis (*p* = 0.90).

### 4.3. Effect on VO_2_ Kinetics

A subject’s oxygen consumption is directly related to the requirements of different exercise intensities and responds directly to the physiological and metabolic demands of peripheral tissues [8,77]. Thus, the ratio of VO_2_ to CO_2_ increases responds via the Respiratory Exchange Ratio (RER) to the change in energy substrates in favor of the priority use of carbohydrates and corresponds (like lactate production) to increased exercise intensity [78]. Different changes in VO_2_ kinetics related to reaction time, ventilatory capacity, or decreased oxygen debt could lead to improved ventilatory efficiency, increased tissue oxygenation at a given intensity, and therefore increased athletic performance [77]. Supplements that induce increased NO production such as beetroot juice may be associated with positive changes in VO_2_ kinetics through multiple mechanisms, including increased vasodilation and a corresponding increase in blood flow to peripheral tissues [25,79].

Along the same lines, Cit could exert its effect on VO_2_ kinetics. In fact, a study that analyzed the impact of its supplementation, using a dose of 6 g/day for 6 days, found significant improvements in MRT and the tissue oxygenation index, which were related to direct improvements in VO_2_ kinetics itself [45]. Similarly, another study by the same group of scientists reported a significant increase in muscle oxygenation at both submaximal and maximal exercise after 16 days of long-term supplementation with 3.4 g/day of Cit [57]. In the same study, however, the improvement in VO_2_ was not significant. Similar results were found by Ashley et al. [56] in MRT and oxygen deficit in a group of 15 active adults who ingested 6 g/day of Cit for 7 days before walking at the submaximal intensity on a treadmill. In the same study, however, the results were not positive for a group of people with an average age of over 70 years old. In contrast, Hickner et al. [43] showed no benefit in VO_2_ peak in their study of physically active men and women. Cutrufello et al. [48] also reported no benefit with respect to VO_2_ max, vasodilation, and VO_2_ at the anaerobic threshold. Along similar lines, another study with similar doses of Cit showed no improvements in VO_2_ max kinetics [49]. The differences in the results shown between the different studies could be explained by the dose and timing of Cit supplementation. As with other NO pathway supplements, chronic intakes of Cit appear to be necessary to obtain greater improvements in the cardiovascular system.

While acute supplementation showed no benefits for VO_2_ kinetic variables, those studies in which chronic supplementation was used (7–16 days) at a dose of approximately 3.4–6 g/day appeared to improve these variables. However, in view of the results obtained in this meta-analysis (*p* = 0.94), it cannot be determined that Cit supplementation has a positive impact on VO_2_. Therefore, further studies should be carried out with chronic doses and with the combination of other potentially synergistic supplements that could show beneficial effects, such as those previously documented in the literature [62].

### 4.4. Effect on Lactate

Lactate is the main metabolite of glycolysis during exercise and its production increases proportionally to the intensity of exercise until it exceeds the mitochondrial clearance capacity and accumulates in the bloodstream [68]. While intensity is the main factor in its increase, the availability of muscle and liver glycogen, exogenous glucose, and the subject’s fiber type, among other factors, determine its production [68]. However, its oxidation is limited by the functionality of the mitochondrion and the associated metabolic flexibility [69]. Although its increase in blood is considered a negative effect, especially at submaximal intensities, the increase in its production is synonymous with increased glycolytic activation and, therefore, more efficient fuel use at higher intensities requiring higher physiological demands [6,69]. Consequently, its increase after a submaximal or maximal test, especially when accompanied by an objective improvement in performance, can mean an increase in exercise intensity and, therefore, in the ability to exercise at a higher intensity [69,80,81]. Various ergo-nutritional aids such as caffeine, beta-alanine, or bicarbonate have been shown to significantly increase lactate levels, accompanied by an improvement in sports performance [82], as well as other NO-related supplements such as beetroot juice [83,84]. Chronic supplementation for 16 days at a dose of 3.4 g/day showed a significant increase in lactate at submaximal and maximal exercise in recreational sportspeople [57]. The same results were observed with other acute supplementation methodologies. Martínez-Sánchez A. et al. [59] found significant increases in lactate after a maximal intermittent test followed by a continuous test with a high intake of Cit (12 g 60 min before the test). Along the same lines, one study showed higher lactate levels after a half-marathon race compared to a placebo, although participants had ingested a much lower dose (3.45 g ingested 2 h before). However, Hickner et al. [43] observed no differences from the control group and between groups when comparing two different doses of Cit (9 g vs. 3 g) in an incremental treadmill test to exhaustion. Although the latter results were not statistically significant, the raw data from the study showed a positive trend, suggesting the effect previously observed in the other studies.

Cit supplementation could increase lactate concentration through an increase in exercise intensity that could be explained by an increased ability to exercise at a higher intensity, with higher bioenergetic efficiency, and with increased tolerance to lactate [20,44]. Although the results of the studies analyzed could be considered positive, the meta-analysis carried out has not shown significant results (*p* = 0.16). These results could be justified by the small number of studies examined. However, there is no consensus from which to conclude that Cit has a positive effect on lactate increase during endurance exercise.

### 4.5. Strengths and Limitations

This systematic review has several strengths and weaknesses. The total number of selected articles represents an important limitation due to the scarcity of studies that have analyzed Cit supplementation on aerobic performance and its variables. Additionally, there has been a broad variety in supplementation protocols, both in dosage and timing, which means that the likelihood of obtaining accurate conclusions is commensurately lower. This may be due to the controversial results of the scientific literature which, for the moment, do not allow for the most appropriate doses and protocols for use to be established. The opportunities for advancement in this area are thus hindered accordingly. Additionally, the variables analyzed and the tests carried out widely differed from one another. Nevertheless, they were all related to the demands of aerobic exercise, and therefore the conclusions have been drawn rigorously.

On a different note, the exhaustive methodology carried out in this systematic review and meta-analysis, in terms of study selection, the analysis of possible biases, the analysis of variables, and the interpretation and quantification of results, is an important strength of the study.

### 4.6. Future Research Lines

According to the results obtained in this study, and considering its strengths and weaknesses, future lines of research should focus on the homogeneous study of Cit supplementation, trying to homogenize doses, timing, and the most representative variables of the effects of supplementation, as is the case with other supplements of the NO pathway.

In view of the promising results, further studies should be proposed that use a chronic supplementation methodology and at higher doses than the usual ones (6 g/min). This would make it possible to measure not only the acute effects on performance but also the possible physiological and metabolic adaptations already analyzed for other supplements [26]. Although this systematic review and meta-analysis has not been able to establish relationships in terms of the sample of studies, it may be of interest to analyze the difference in effects between participants of different levels, due to the differences already reported in similar supplements between elite and amateur athletes [24]. Additionally, the pharmacokinetics of CWB and the absorption mechanisms of exogenous supplementation are two areas to be studied in depth that could provide new and more specific knowledge to extend the understanding of the effects of Cit supplementation.

Finally, and considering the role of Cit in the NO metabolic pathways and the close relationship with the Arg on the one hand, and nitrate and nitrite on the other, it could be of great interest to analyze the joint effects of different supplements of this pathway on sports performance, as well as on different associated physiological and metabolic variables.

## 5. Conclusions

This systematic review and meta-analysis has shown no significant positive effects of Cit supplementation on the different variables analyzed in relation to aerobic sports performance. Chronic supplementation (6–16 days) with an amount between 3.4 and 6 g/day showed a positive tendency to improve aerobic performance, but without statistical significance. Acute supplementation (1–3 h) with 1–12 g of Cit did not significantly improve aerobic performance or any related variables. No improvement in VO_2_ kinetics or the RPE was observed after Cit supplementation, irrespective of doses and timing. In contrast, Cit showed a positive tendency to increase lactate but with no significant results. Cit supplementation does not appear to be effective in increasing aerobic athletic performance and is therefore not recommended for this purpose. Due to the paucity of studies on this subject, as well as the heterogeneity of the protocols used, further studies are needed to understand the impact of such supplementation on athletic performance.

## Figures and Tables

**Figure 1 nutrients-14-03479-f001:**
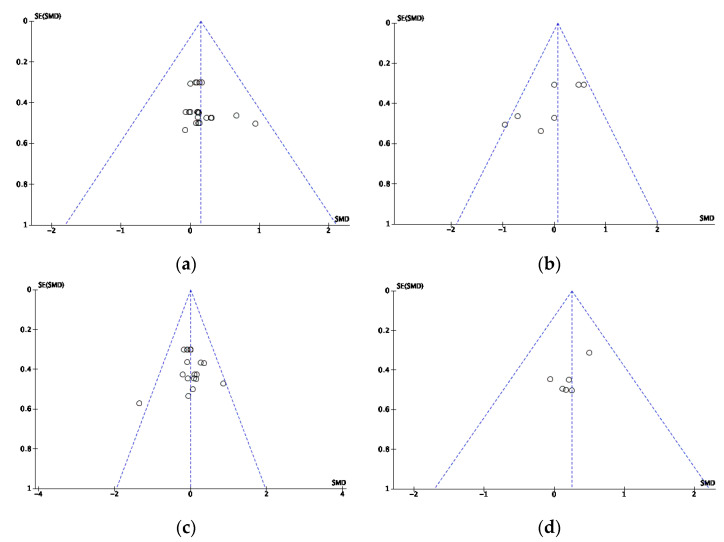
Funnel plots of the standard error of aerobic (**a**) the RPE, (**b**) VO_2_ kinetics, (**c**) lactate, and (**d**) sports performance data, using Hedges’ g. SE: standard error; SMD: standard mean difference.

**Figure 2 nutrients-14-03479-f002:**
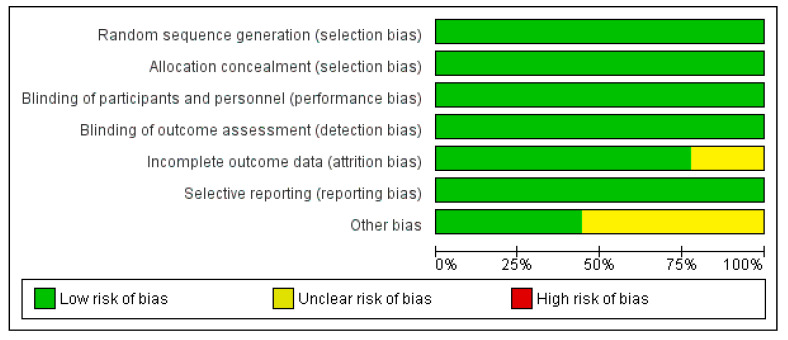
Risk-of-bias graph expressed as percentages.

**Figure 3 nutrients-14-03479-f003:**
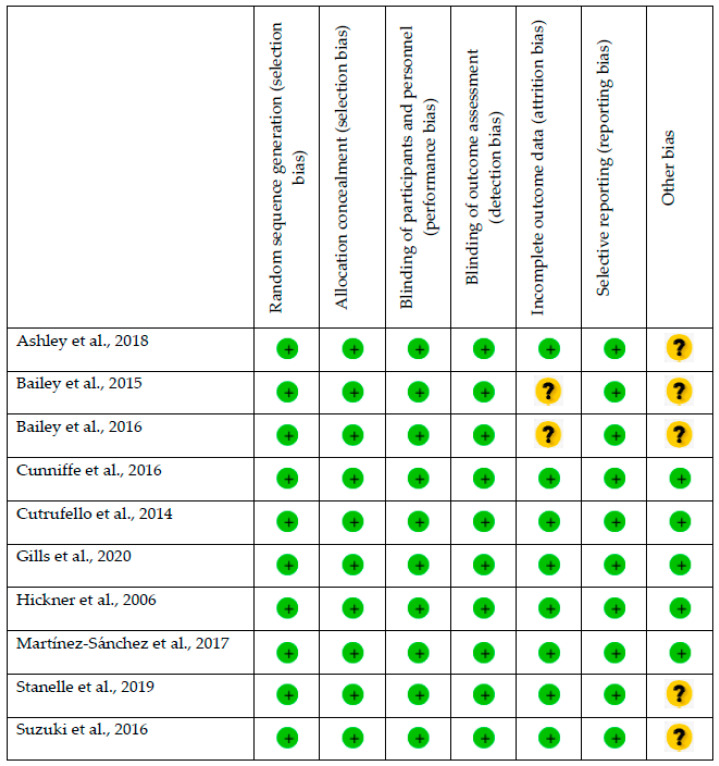
Summary of the risk of bias. Review of the authors’ j of the risk of bias in the different items selected from each article included in the systematic review and meta-analysis. 
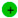
 indicates a low risk of bias; 
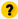
 indicates an unknown risk of bias [43,44,45,48,49,56,57,58,59,60].

**Figure 4 nutrients-14-03479-f004:**
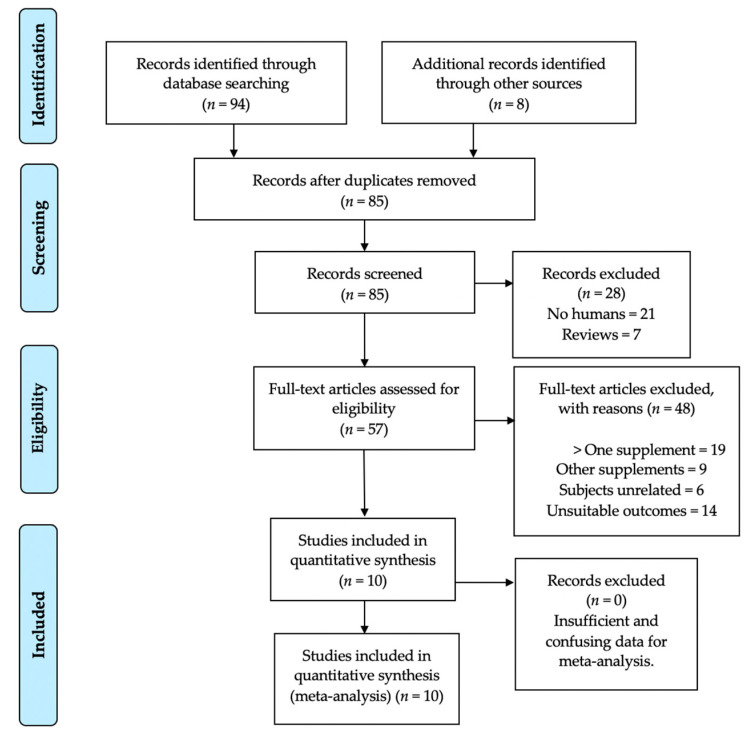
Flowchart of the process of selection, screening, suitability, and inclusion of articles included in the systematic review and meta-analysis. Adapted from PRISMA guidelines [50].

**Figure 5 nutrients-14-03479-f005:**
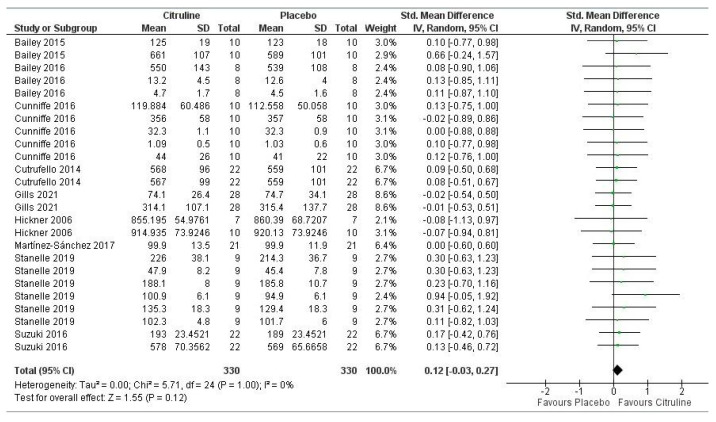
Forest plot comparative study of the effects of Cit supplementation on aerobic sports performance (≤VO_2_ max.) [43,44,45,48,49,57,58,59,60].

**Figure 6 nutrients-14-03479-f006:**
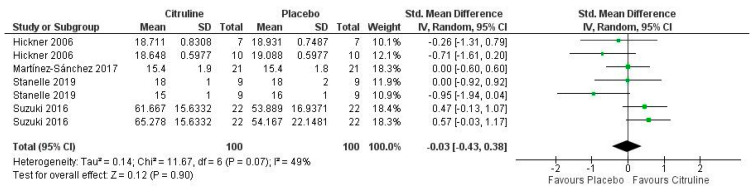
Forest plot comparative study of the effects of Cit supplementation on the RPE [43,49,59,60].

**Figure 7 nutrients-14-03479-f007:**
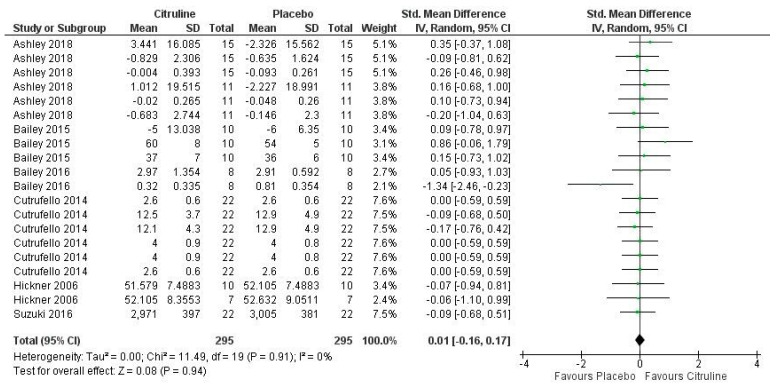
Forest plot comparative study of the effects of Cit supplementation on VO_2_ kinetics [43,45,48,49,56,57].

**Figure 8 nutrients-14-03479-f008:**
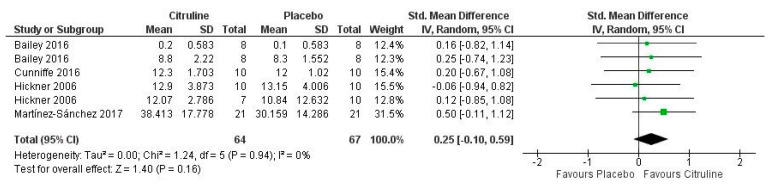
Forest plot comparative study of the effects of Cit supplementation on lactate [43,48,57,59].

**Table 1 nutrients-14-03479-t001:** Design, characteristics, and gender of subjects. Type, dose, and timing of supplementation of the studies that were included in the systematic review and meta-analysis.

Study design	Randomized, double-blind, crossover, and placebo-controlled		10 studies [43,44,45,48,49,56,57,58,59,60]
Characteristics of the subjects	Recreational		4 studies [45,57,58,59]
	Intercollegiate		1 study [48]
	Well-trained		3 studies [44,49,60]
	Active		2 studies [43,56]
Gender of subjects	Male		7 studies [44,45,49,57,58,59,60]
	Female		0 studies
	Both		3 studies [43,48,56]
Type of Citrulline supplementation	L-Citrulline		6 studies [44,45,49,57,59,60]
	Watermelon Juice		1 study [48]
	Watermelon Juice + L-Citrulline		2 studies [57,59]
	Citrulline Malate		2 studies [44,58]
Dose used	1 g of supplement		1 study [48]
	2.4 g of supplement		1 study [49]
	3 g of supplement		1 study [43]
	3.45 g of supplement		2 studies [57,59]
	6 g of supplement		4 studies [45,48,56,60]
	8 g of supplement		1 study [58]
	9 g of supplement		1 study [43]
	12 g of supplement		1 study [44]
Intake time	Acute	1 h before the test	2 study [44,58]
		1–2 h before the test	1 study [48]
		2 h before	1 study [59]
		3 h before	1 study [43]
	Chronic	1 day	1 study [43]
		6 days	1 study [45]
		7 days	3 studies [49,56,60]
		16 days	1 study [57]

**Table 2 nutrients-14-03479-t002:** Summary of studies included in the systematic review and meta-analysis investigating the effect of Cit on aerobic sports performance (≤VO_2_ max).

Author/s	Population	Intervention	Tests	Variables Analyzed	Main Conclusions
Bailey et al., 2015 [45]	Ten healthy, active, recreational sportspeople. Men (19 ± 1 years).	Randomized, double-blind, crossover, and placebo-controlled; 6 g Citrulline + 4.3 g maltodextrin (with 500 mL water) for 6 days + 90 min before the test.	Two cycle-ergometer tests: 1 moderate intensity test on day 6 (90% of GET) + 1 high-intensity test on day 7 (GET + difference between GET and VO_2_ max work).	Total work completed. Exercise tolerance time.	↑ ↑
Bailey et al., 2016 [57]	Eight healthy, active, recreational sportspeople. Men (22 ± 2 years).	Randomized, double-blind, crossover, and placebo-controlled; ~3.4 g/day of L-Citrulline in 300 mL of Watermelon Juice Concentrate for 16 days.	Constant cycle-ergometer work at 70% of VO_2_ max to exhaustion.	Time to exhaustion.	↔
			Six min of constant work (70% VO_2_ max.) + 30 s all-out on cycle-ergometer.	Total work in 30“ Total work in 10”	↔ ↑
Cunniffe et al., 2016 [44]	Ten well-trained, healthy men (23.5 ± 3.7 years).	Randomized, double-blind, crossover, placebo-controlled; 12 g Citrulline Malate (dissolved in 400 mL of water) 60 min before completing the tests.	Two exercises of 10 repetitions of 15 s all-out intensity (sprint) on a cycle-ergometer, followed by 5 min rest and an incremental test to exhaustion at 100% of individual peak power.	Time to exhaustion. Average Power (W) Average speed Distance traveled Work (KJ)	↔ ↔ ↔ ↔ ↔
Cutrufello et al., 2014 [48]	Eleven healthy men (20.6 ± 1.2 years) and 11 healthy women (21.0 ± 1.3 years); intercollegiate team players.	Randomized, double-blind, crossover, placebo-controlled; 6 g L-Citrulline in 710 mL sucrose solution ingested 1 h or 2 h before the test.	Bruce’s treadmill protocol to exhaustion.	Time to exhaustion group 1	↔
		Randomized, double-blind, crossover, placebo-controlled; 710 mL of watermelon juice (1 g of L-Citrulline) taken 1 or 2 h before the test.	Bruce’s treadmill protocol to exhaustion.	Time to exhaustion group 1	↔
Hickner et al., 2006 [43]	Seventeen physically active men and women (18–40 years).	Randomized, double-blind, crossover, and placebo-controlled; 3 × 3 g of L-Citrulline for 24 h.	Incremental treadmill test to exhaustion.	Time to exhaustion group 1	↑
	Ten physically active men and women (18–40 years).	Randomized, double-blind, crossover, and placebo-controlled; 3 g L-Citrulline 3 h before the test.	Incremental treadmill test to exhaustion.	Time to exhaustion group 2	↑
Martínez-Sánchez et al., 2017 [59]	Twenty-two healthy men; amateur runners (35.3 ± 11.4 years).	Randomized, double-blind, crossover, and placebo-controlled; 500 mL of Fashion Watermelon Juice enriched with L-Citrulline (3.45 g per 500 mL) taken 2 h before the race.	Half-Marathon Race	Time in the Half-Marathon	↔
Stanelle et al., 2019 [60]	Nine well-trained male triathletes or cyclists and competitors (24 ± 3 years).	Randomized, double-blind, crossover, and placebo-controlled; 6 g/day of Citrulline or 7 days + one dose of 6 g 2h before the test.	Forty km individual time trial on cycle-ergometer.	Time to complete the test Average power Average moment of force Average cadence	↑ ↑ ↔ ↔
			SRT: 6 sprints of 1 minute at 120% of maximum power on the cycle-ergometer immediately after the time trial.	Average moment of force Average cadence	↔ ↔
Gills et al., 2020 [58]	Twenty-eight trained men (20.9 ± 2.8 years).	Randomized, double-blind, crossover, and placebo-controlled; 8 g/day of Citrulline Malate 1 h before the test.	Exhaustion test at 90% of VO_2_ peak above 40 rpm.	Time to exhaustion Total work completed	↔ ↔

↑: Statistically significant improvement of Citrulline treatment; ↔: no statistically significant improvement of Citrulline treatment. W: watts (power). KJ: Kilojoules.

**Table 3 nutrients-14-03479-t003:** Summary of studies included in the systematic review and meta-analysis investigating the effect of Cit on the rate of perceived exertion (RPE).

Author/s	Population	Intervention	Test	Variables Analyzed	Main Conclusions
Hickner et al., 2006 [43]	Seven physically active men and women (18–40 years).	Randomized, double-blind, crossover, and placebo-controlled; 3 × 3 g of L-Citrulline for 24 hrs.	Incremental treadmill test to exhaustion.	Submaximal RPE	↓
	Ten physically active men and women (18–40 years).	Randomized, double-blind, crossover, and placebo-controlled; 3 g L-Citrulline 3 h before the test.	Incremental treadmill test to exhaustion.	Submaximal RPE	↓
Martínez-Sánchez et al., 2017 [59]	Twenty-two healthy male amateur runners (35.3 ± 11.4 years).	Randomized, double-blind, crossover, and placebo-controlled; 500 mL of Fashion Watermelon Juice enriched with L-Citrulline (3.45 g per 500 mL) taken 2 h before the race.	Half-Marathon Race	RPE	↔
Stanelle et al. (2019) [60]	Nine well-trained male triathletes or cyclists and competitors (24 ± 3 years).	Randomized, double-blind, crossover, and placebo-controlled; 6 g/day of L-Citrulline for 7 days + a dose of 6 g 2 h before the test.	Forty km individual time trial on cycle-ergometer.	RPE	↓
			SRT: 6 sprints of 1 min at 120% of maximum power on the cycle-ergometer immediately after the time trial.	RPE	↑
Suzuki et al., 2016 [49]	Twenty-two healthy and well-trained men (29 ± 8.4 years)	Randomized, double-blind, crossover, and placebo-controlled; 2.4 g/day of L-Citrulline (9 capsules) for 1 week before the test (before bedtime) + 1 h before the test.	Four km individual time trial on cycle-ergometer.	RPE	↔

↑: Statistically significant improvement from Citrulline treatment; ↔ no statistically significant improvement from Citrulline treatment. ↓: statistically significant worsening from Citrulline treatment. RPE: Rate of Perceived Exertion.

**Table 4 nutrients-14-03479-t004:** Summary of studies included in the systematic review and meta-analysis investigating the effect of IAC on oxygen uptake kinetics (VO_2_).

Author/s	Population	Intervention	Test	Variables Analyzed	Main Conclusions
Ashley et al., 2018 [56]	Fifteen active adults (8 women, 7 men) (22 ± 2 years).	Randomized, double-blind, crossover, placebo-controlled; 6 g/day L-Citrulline for 7 days.	Treadmill running at 40% HRR. Measurement of oxygen consumption by indirect calorimetry.	Net oxygen expenditure group 1 Mean Response Time (MRT) group 1 Oxygen deficiency group 1	↔ ↑ ↑
	Eleven active adults (7 women, 4 men) (74 ± 7 years).			Net oxygen expenditure group 2 Mean Response Time (MRT) group 2 Oxygen deficiency group 2	↔ ↔ ↔
Bailey et al., 2015 [45]	Ten healthy, active, recreational subjects. Men (19 ± 1 years).	Randomized, double-blind, crossover, placebo-controlled; 6 g L-Citrulline + 4.3 g maltodextrin (with 500 mL water) for 6 days + 90 min before the test.	Two cycle-ergometer tests: 1 moderate intensity test on day 6 (90% of GET) + 1 high-intensity test on day 7 (GET + difference between GET and VO_2_ max work).	Pulmonary VO_2_ Mean Response Time (s) Mean Response Time (MRT) group 2. Tissue oxygenation index (%)	↑ ↑ ↑
Bailey et al., 2016 [57]	Eight healthy, active, recreational subjects. Men (22 ± 2 years).	Randomized, double-blind, crossover, placebo-controlled. ~3.4 g/day of L-Citrulline in 300 mL of Watermelon Juice Concentrate for 16 days.	Constant cycle-ergometer works at 70% of VO_2_ max to exhaustion.	Muscle oxygenation Pulmonary VO_2_	↑ ↔
			Six min of constant work (70% VO_2_ max) + 30′ all-out on cycle-ergometer.	Muscle oxygenation PulmonaryVO_2_	↑ ↔
Cutrufello et al., 2014 [48]	Eleven healthy men (20.6 ± 1.2 years) and eleven healthy women (21.0 ± 1.3 years); intercollegiate team players.	Randomized, double-blind, crossover, placebo-controlled; 6 g L-Citrulline in 710 mL sucrose solution ingested 1 or 2 h before the test.	Bruce’s treadmill protocol to exhaustion.	VO_2_ max. group 1 Vasodilatation group 1 Anaerobic threshold defined by gas exchange group 1	↔ ↔ ↔
		Randomized, double-blind, crossover, placebo-controlled; 710 mL of Watermelon Juice (1 g L-Citrulline) ingested 1 or 2 h before the test.	Bruce’s treadmill protocol to exhaustion.	VO_2_ max. group 2 Vasodilatation group 2 Anaerobic threshold defined by gas exchange group 2	↔ ↔ ↔
Hickner et al., 2006 [43]	Seven physically active men and women (18–40 years).	Randomized, double-blind, crossover, and placebo-controlled; 3 x 3 g of L-Citrulline for 24 hrs.	Incremental treadmill test to exhaustion.	VO_2_ peak group 1	↔
	Ten physically active men and women (18–40 years).	Randomized, double-blind, crossover, and placebo-controlled; 3 g L-Citrulline 3 h before the test.	Incremental treadmill test to exhaustion.	VO_2_ peak group 2	↔
Suzuki et al., 2016 [49]	Twenty-two healthy and well-trained men (29 ± 8.4 years).	Randomized, double-blind, crossover, and placebo-controlled; 2.4 g/day of L-Citrulline (9 capsules) for 1 week before the test (before bedtime) + 1 h before the test.	Four km individual time trial on cycle-ergometer.	VO_2_ kinetics	↔

↑: Statistically significant improvement of Citrulline treatment; ↔ no statistically significant improvement of Citrulline treatment. VO_2_: Oxygen uptake. VO_2_ max: Maximal Oxygen uptake. HRR: Heart Rate Reserve. MRT: Mean Response Time. GET Gas Exchange Threshold.

**Table 5 nutrients-14-03479-t005:** Summary of studies included in the systematic review and meta-analysis investigating the effect of Cit on lactate.

Author/s	Population	Intervention	Test	Variables Analyzed	Main Conclusions
Bailey et al., 2016 [57]	Eight healthy, active, recreational subjects. Men (22 ± 2 years).	Randomized, double-blind, crossover, placebo-controlled; ~3.4 g/day of L-Citrulline in 300 mL of Watermelon Juice Concentrate for 16 days.	Constant cycle-ergometer works at 70% of VO_2_ max to exhaustion.	Lactate	↑
			Six min of constant work (70% VO_2_ max) + 30” all-out on cycle-ergometer.	Lactate	↑
Cunniffe et al.,2016 [44]	Ten well-trained healthy men (23.5 ± 3.7 years).	Randomized, double-blind, crossover, placebo-controlled; 12 g Citrulline Malate (dissolved in 400 mL water) 60 min before completing the tests.	Two exercises of 10 repetitions of 15 s all-out (sprint) on a cycle-ergometer, followed by 5 min of rest and an incremental test to exhaustion at 100% of individual peak power.	Lactate	↑
Hickner et al., 2006 [43]	Seven physically active men and women (18–40 years).	Randomized, double-blind, crossover, placebo-controlled; 3 × 3 g of L-Citrulline for 24 h	Incremental treadmill test to exhaustion.	Lactate	↔
	Ten physically active men and women (18–40 years).	Randomized, double-blind, crossover, placebo-controlled; 3 g L-Citrulline 3 h before the test.	Incremental treadmill test to exhaustion.	Lactate	↔
Martínez-Sánchez et al., 2017 [59]	Twenty-two healthy male amateur runners (35.3 ± 11.4 years).	Randomized, double-blind, crossover, placebo-controlled; 500 mL of Fashion Watermelon Juice enriched with L-Citrulline (3.45 g per 500 mL) taken 2 h before the race.	Half-Marathon Race	Lactate	↑

↑: Statistically significant improvement of Citrulline treatment; ↔ no statistically significant improvement of Citrulline treatment.

## Data Availability

Not applicable.

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
