# Peer review of "Effects of Citrulline Supplementation on Different Aerobic Exercise Performance Outcomes: A Systematic Review and Meta-Analysis"

_nutrients, 2022, doi:10.3390/nu14173479_

Round 1

Reviewer 1 Report

Comments for “Effects of Citrulline Supplementation on Different Aerobic Exercise Performance Outcomes: A Systematic Review and Meta-Analysis

The purpose of this meta-analysis was to examine the citrulline (Cit) supplementation on aerobic performance, lactate, VO2, and RPE. A total of 9 studies were included, and the analyses showed that Cit supplementation did not have any benefits for aerobic exercise performance and related outcomes. Overall, the meta-analysis was well-conducted, and the authors did a fine job presenting the results and writing the article. I think the current form of the article is meeting the publishing standards of the journal. Therefore, I do not have any specific comments. The only thing I recommend is to proof read the entire manuscript, as there are still a few minor language errors need to be corrected.

Author Response

Point-by-Point Response to Reviewer’s Comments

We would like to sincerely thank reviewer 1 for their helpful recommendations. We feel that the quality of the manuscript has been significantly improved with these modifications and improvements based on the reviewers’ suggestions and comments. We hope our revision will lead to an acceptance of our manuscript for publication in Nutrients.

In advance,

Kind regards

REVIEWER: The purpose of this meta-analysis was to examine the citrulline (Cit) supplementation on aerobic performance, lactate, VO2, and RPE. A total of 9 studies were included, and the analyses showed that Cit supplementation did not have any benefits for aerobic exercise performance and related outcomes. Overall, the meta-analysis was well-conducted, and the authors did a fine job presenting the results and writing the article. I think the current form of the article is meeting the publishing standards of the journal. Therefore, I do not have any specific comments. The only thing I recommend is to proof read the entire manuscript, as there are still a few minor language errors need to be corrected.

AUTHORS: Thank you for your comments and recommendations. Although the manuscript was reviewed by a native English speaker prior to its initial submission, the manuscript has been revised again in order to improve possible errata.

Reviewer 2 Report

The focus of this systematic review and meta-analysis is the role of Citrulline Supplementation on aerobic exercise performance and related outcomes. 

Abstract: Abstract reported a total of 9 studies included in the analysis but in the text of this systematic review and meta-analysis are 10

Introduction: Introduction provides sufficient background

Methods: The methods of the study is adequately described

In the section ”Study Selection” why reported Figure 6?

In the section “Publication Bias” why reported Figure 1?

In the section”Assessing the quality of investigations why reported Figure 2 and Figure 3?

Results: The results are not clearly presented.

The section Results could be strutturated in

3.1. Main search -Studies selection;

3.2. Study Characteristics and Quality Assessment;

3.2A Citrulline on aerobic performance, on RPE, on VO2 kinetics and on lactate).

3.2B Quality Assessment

3.3 Meta-analysis; assessed the impact of Cit supplementation on any benefits for aerobic exercise performance and specific outcomes considered (aerobic performance, on RPE, on VO2 kinetics and on lactate).

3.3A Meta-analysis of Citrulline on Aerobic Performance (≤VO2 max.);

3.3B Meta-analysis of Citrulline on the RPE;

3.3C Meta-analysis of Citrulline on VO2 kinetics;

3.3D Meta-analysis of Citrulline on Lactate

3.4 Sensitivity analysis;

3.5 Pubblication bias

The studies included in the Systematic Review and Meta-Analysis are 10 but in the results were reported only 9 references (43–45,48,49,56–58).

In results they reported “studies used a chronic supplementation methodology ranging from 1 day to 16 days (43,45,56,57,59,62)but the study of reference number 62 is not included in the Systematic Review and Meta-Analysis.

To better understand the effects of Citrulline Supplementation on Different Aerobic Exercise Performance it is important to investigate type of Citrulline Supplementation and timing of ingestion. Please estimate the results (aerobic exercise performance , aerobic performance, on RPE, on VO2 kinetics and on lactate) based on Type of Citrulline supplementation (L-Citrulline, Watermelon Juice + L-Citrulline, Citrulline-Malate) and Intake time (Acute or Chronic). For each outcome must be reported number of studies included, combined pooled SMD (Value 95% CI and p), Test of heterogeneity (Q, I2% and p) and Publication bias (Egger test)

Discussion:

Synergistic effects between different supplements is an exclusion criteria in your Systematic Review and Meta-Analysis studies. Please removed lines 281-283 “Finally, as has been previously documented in the literature (62,85), while the area related to the synergistic effects between different supplements remains unexplored, it could have enormous potential in the associated effects” and relative references.

Synergistic effects between different supplements not remains unexplored but in your Systematic Review and Meta-Analysis studies in which supplementation did not involve Citrulline alone were discarded.

References:

Update bibliographical reference 50 and removed reference number 62 and 85

Conclusion:

The conclusions are consistent with the evidence and arguments presented and they address the question posed.

FIGURE:

Figure 1 “Funnel plots of the standard error of aerobic (a) RPE, (b) VO2 kinetics, (c) lactate, and (d) sports performance data, using Hedges' g. SE: standard error; SMD: standardised mean difference” must be moved in the section 3.5 Pubblication bias of Results.

Figure 2 “Risk of bias: review of the authors' judgement of the risk of bias in the different items of the articles included in the systematic review and meta-analysis, presented in percentages according to their appropriateness” must be shifted in the section of 3.2B Quality Assessment of Results.

Figure 3 “Summary of risk of bias. Review of the authors' judgement of the risk of bias in the different items selected from each article included in the systematic review and meta-analysis. indicates low risk of bias; indicates unknown risk of bias; indicates high risk of bias” must be shifted in the section of 3.2B Quality Assessment of Results.

Insert bibliographical references in the first column, after author and year.

Figure 4 ”Flowchart of the process of selection, screening, suitability, and inclusion of articles included in the systematic review and meta-analysis. Adapted from (50)” become FIGURE 1

This figure will be in in the section 3.1. Main search -Studies selection of Results.

The studies included in the Systematic Review and Meta-Analysis are 10 but Prisma flowchart reported 9 studies. Correct, please .

The text in the boxes of Prisma flowchart are cutted. Please done readable

Update bibliographical reference 50 of Prisma “Page M J, McKenzie J E, Bossuyt P M, Boutron I, Hoffmann T C, Mulrow C D et al. The PRISMA 2020 statement: an updated guideline for reporting systematic reviews BMJ 2021; 372:n71 doi:10.1136/bmj.n71”

Figure 5: Forest plot comparative study of the effects of Cit supplementation on aerobic sports performance (≤VO2 max.).  This figure will be in in the section 3.3A Meta-analysis of Citrulline on Aerobic Performance (VO2 max.) of Results.

Figure 6:Forest plot comparative study of the effects of Cit supplementation on the RPE.  This figure will be in in the section 3.3B Meta-analysis of Citrulline on the RPE of Results

Figure 7:Forest plot comparative study of the effects of Cit supplementation on VO2 kinetics. This figure will be in in the section 3.3C Meta-analysis of Citrulline on VO2 kinetics of Results

Figure 8:Forest plot comparative study of the effects of Cit supplementation on lactate.  This figure will be in in the section 3.3D Meta-analysis of Citrulline on Lactate of Results.

TABLE

Table 1 “Design, characteristics, and gender of subjects. Type, dose, and timing of supple-mentation of the studies included in the systematic review and meta-analysis” must be moved in the Supplementary Materials.

Table 2: Removed

Table 3: Removed

Table 4: Removed

Table 5: Removed

Create a new Tables (Table 1 and Table 2)

Table 1. Summary of studies included in the systematic review and meta-analysis investigating the effect of Citrulline Supplementation on Different Aerobic Exercise Performance Outcomes

Create a new Table.

The first four columns of table 2, 3, 4 and 5 are the same.

Crate a new summary table, named table 1 with the first 4 columns of table 2, 3, 4 and 5, the different variables analysed and main conclusion reported in the tables 2-5.

Check population of reference number 43. There are different number in the tables 2-5.

This table will be in the section 3.2. Study Characteristics and Quality Assessment of Results.

Table 2: Meta-analysis of Cit supplementation and any benefits for aerobic exercise performance and specific outcomes considered (aerobic performance, on RPE, on VO2 kinetics and on lactate).

Create a new Table 2

For each outcome must be reported number of studies included, combined pooled SMD (Value 95% CI and p), Test of heterogeneity (Q, I2% and p) and Publication bias (Egger test)

The table 2 will be shows the results of the meta-analysis that assessed the impact of Cit supplementation on any benefits for aerobic exercise performance and specific outcomes considered (aerobic performance, on RPE, on VO2 kinetics and on lactate).

The aerobic exercise performance is the combination of aerobic performance, on RPE, on VO2 kinetics and on lactate.

Considering the characteristics of the studies, insert results of stratified analysis of the single outcome (aerobic exercise performance , aerobic performance, on RPE, on VO2 kinetics and on lactate) estimates according to the Characteristics of the subjects (Recreational, Well trained, Active condition); Gender of subjects; Type of Citrulline supplementation (L-Citrulline, Watermelon Juice + L-Citrulline, Citrulline-Malate) and Intake time (Acute or Chronic).

Some items need to be addressed prior to publication of the study. The results are not clearly presented and some figures presented in the methods are results. I recommend the publication of present research after majorr revisions and done more readable the paper.

Author Response

Point-by-Point Response to Reviewer’s Comments

The authors want to thank the comments and suggestions done by Reviewer 2 to improve the manuscript.  Each comment will be replied to below (colored in yellow) and the authors will do corresponding changes in the article document.

However, major proposed changes are due to structure and methods or results presentation. In this sense, the structure proposed by the reviewer seems adequate to show the results of the meta-analysis. However, although we have modified some headings in the results section following the reviewer's suggestions and reorganized the results structure for systematic review and meta-analysis, the authors would like to maintain the aspects related to the quality of the articles included in the methods section as we have already published in other meta-analyses of our research group. Our research group has already published another 5 Systematic Reviews and Meta-Analysis with the same structure as in the manuscript. Indeed, some of them have been published in this Journal (Nutrients).

In this way, the authors consider that everything related to the bias of the articles included in this meta-analysis should be included in the material and methods section. This is because, on the one hand, they are not results that respond to the objective of the manuscript, and, on the other hand, these data represent the way in which we chose and validated each of the articles included. That is, what methods we used to include the articles in the meta-analysis.

In addition, in the presentation of results, we believe it is important to order them according to the outcomes studied so that the reader can interpret them individually. We know that much of the information is repeated between the tables, but in this way, the authors believe that it is much more intuitive for the reader since he/she has all the information in the same table and does not have to go to other tables to look for information related to each study.

Nevertheless, if in spite of the changes made, reviewer 2 considers the change of structure that he/she proposes to us to be indisputable, we would have no problem in resolving it in order to facilitate its publication.

In advance,

Kind regards

Abstract

REVIEWER: Abstract reported a total of 9 studies included in the analysis but in the text of this systematic review and meta-analysis are 10

AUTHORS: Thanks for your observation. The authors have changed it.

Introduction

REVIEWER: Introduction provides sufficient background

AUTHORS: The authors thank your comment.

Methods

REVIEWER: The methods of the study is adequately described

AUTHORS: Thank you for your comment.

REVIEWER: In the section ”Study Selection” why reported Figure 6?

AUTHORS: Thank you. Figure 6 has been replaced to figure 4.

REVIEWER: In the section “Publication Bias” why reported Figure 1?

AUTHORS: Thank you for your question. Figure 1 shows funnel plots of the standard error of each outcome. Performing Egger's statistical test detects publication bias, considering p ≤ 0.05.

REVIEWER: In the section”Assessing the quality of investigations why reported Figure 2 and Figure 3?

AUTHORS: Thank you for your question. Both figures represent the risk of publication bias, and therefore the quality of investigations. Concretely, figure 2 is the risk of bias graph expressed as percentages, and figure 3 represents the summary of all risk of bias items

Results:

REVIEWER: The results are not clearly presented.
The section Results could be strutturated in
3.1. Main search -Studies selection;
3.2. Study Characteristics and Quality Assessment;

3.2A Citrulline on aerobic performance, on RPE, on VO2 kinetics and on lactate).

3.2B Quality Assessment
3.3 Meta-analysis; assessed the impact of Cit supplementation on any benefits for aerobic exercise performance and specific outcomes considered (aerobic performance, on RPE, on VO2 kinetics and on lactate).

3.3A Meta-analysis of Citrulline on Aerobic Performance (≤VO2 max.); 3.3B Meta-analysis of Citrulline on the RPE;
3.3C Meta-analysis of Citrulline on VO2 kinetics;
3.3D Meta-analysis of Citrulline on Lactate

3.4 Sensitivity analysis; 3.5 Pubblication bias

AUTHORS: Thank you very much for your suggestion. Indeed, the structure proposed by the reviewer seems adequate to show the results of the meta-analysis. However, although we have modified some headings in the results section following the reviewer's suggestions and reorganized the results structure for systematic review and meta-analysis, the authors would like to maintain the aspects related to the quality of the articles included in the methods section as we have already published in other meta-analyses of our research group as explained to the reviewer at the beginning of this letter.

Viribay, A., Burgos, J., Fernández-Landa, J., Seco-Calvo, J., & Mielgo-Ayuso, J. (2020). Effects of arginine supplementation on athletic performance based on energy metabolism: A systematic review and meta-analysis. Nutrients, 12(5), 1300.

Mielgo-Ayuso, J., Calleja-Gonzalez, J., Marqués-Jiménez, D., Caballero-García, A., Córdova, A., & Fernández-Lázaro, D. (2019). Effects of creatine supplementation on athletic performance in soccer players: a systematic review and meta-analysis. Nutrients, 11(4), 757.

Castañeda-Babarro, A., Marqués-Jiménez, D., Calleja-González, J., Viribay, A., León-Guereño, P., & Mielgo-Ayuso, J. (2020). Effect of listening to music on Wingate anaerobic test performance. A systematic review and meta-analysis. International Journal of Environmental Research and Public Health, 17(12), 4564.

Santibañez-Gutierrez, A., Fernández-Landa, J., Calleja-González, J., Delextrat, A., & Mielgo-Ayuso, J. (2022). Effects of Probiotic Supplementation on Exercise with Predominance of Aerobic Metabolism in Trained Population: A Systematic Review, Meta-Analysis and Meta-Regression. Nutrients, 14(3), 622.

Fernández-Lázaro, D., Gallego-Gallego, D., Corchete, L. A., Fernández Zoppino, D., González-Bernal, J. J., García Gómez, B., & Mielgo-Ayuso, J. (2021). Inspiratory muscle training program using the powerbreath®: Does it have ergogenic potential for respiratory and/or athletic performance? a systematic review with meta-analysis. International Journal of Environmental Research and Public Health, 18(13), 6703.

REVIEWER: The studies included in the Systematic Review and Meta-Analysis are 10 but in the results were reported only 9 references (43–45,48,49,56–58).

AUTHORS: Thanks for your comment. The authors have included the missing reference.

REVIEWER: In results they reported “studies used a chronic supplementation methodology ranging from 1 day to 16 days (43,45,56,57,59,62)” but the study of reference number 62 is not included in the Systematic Review and Meta-Analysis.

AUTHORS: Thanks for the correction. The authors have modified the reference.

REVIEWER: To better understand the effects of Citrulline Supplementation on Different Aerobic Exercise Performance it is important to investigate type of Citrulline Supplementation and timing of ingestion. Please estimate the results (aerobic exercise performance , aerobic performance, on RPE, on VO2 kinetics and on lactate) based on Type of Citrulline supplementation (L-Citrulline, Watermelon Juice + L-Citrulline, Citrulline-Malate) and Intake time (Acute or Chronic).

For each outcome must be reported
number of studies included, combined pooled SMD (Value 95% CI and p), Test of heterogeneity (Q, I2% and p) and Publication bias (Egger test)

AUTHORS: Thank you very much for your suggestion. Although the idea is very accurate, due to the scarce studies on some of the forms of citrulline administration (1 or 2) it is very difficult to make a meta-analysis.

Discussion:

REVIEWER: Synergistic effects between different supplements is an exclusion criteria in your Systematic Review and Meta-Analysis studies. Please removed lines 281-283 “Finally, as has been previously documented in the literature (62,85), while the area related to the synergistic effects between different supplements remains unexplored, it could have enormous potential in the associated effects” and relative references.

AUTHORS: Thanks for your comment. The authors have deleted mentioned lines and references.

REVIEWER: Synergistic effects between different supplements do not remain unexplored but in your Systematic Review and Meta-Analysis studies in which supplementation did not involve Citrulline alone were discarded.

AUTHORS: Authors understand the Reviewer´s point and agree with it.

References:

REVIEWER: Update bibliographical reference 50

AUTHORS: Thanks for the correction. The reference has been updated.

REVIEWER: Removed reference number 62 and removed reference number 85

AUTHORS: Thanks for your comment. Reference 85 has been removed and 62 still remains as it is used in 4.3 section in discussion.

Conclusion:

REVIEWER: The conclusions are consistent with the evidence and arguments presented and they address the question posed.

AUTHORS: Thank you for your comment.

FIGURE:

AUTHORS: Thank you for your suggestions about structure organization. As we mentioned at the beginning of this letter, we would like to be able to maintain the current structure. If this is not possible, please do not hesitate to let us know so that we can change it.

REVIEWER: Figure 1 “Funnel plots of the standard error of aerobic (a) RPE, (b) VO2 kinetics, (c) lactate, and (d) sports performance data, using Hedges' g. SE: standard error; SMD: standardised mean difference” must be moved in the section 3.5 Pubblication bias of Results.

REVIEWER: Figure 2 “Risk of bias: review of the authors' judgement of the risk of bias in the different items of the articles included in the systematic review and meta-analysis, presented in percentages according to their appropriateness” must be shifted in the section of 3.2B Quality Assessment of Results.

REVIEWER: Figure 3 “Summary of risk of bias. Review of the authors' judgement of the risk of bias in the different items selected from each article included in the systematic review and meta-analysis. indicates low risk of bias; indicates unknown risk of bias; indicates high risk of bias” must be shifted in the section of 3.2B Quality Assessment of Results.

Insert bibliographical references in the first column, after author and year.

REVIEWER: Figure 4 ”Flowchart of the process of selection, screening, suitability, and inclusion of articles included in the systematic review and meta-analysis. Adapted from (50)” become FIGURE 1

REVIEWER: This figure will be in in the section 3.1. Main search -Studies selection of Results.

REVIEWER: The studies included in the Systematic Review and Meta-Analysis are 10 but Prisma flowchart reported 9 studies. Correct, please.

REVIEWER: The text in the boxes of Prisma flowchart are cutted. Please done readable
Update bibliographical reference 50 of Prisma “Page M J, McKenzie J E, Bossuyt P M, Boutron I, Hoffmann T C, Mulrow C D et al. The PRISMA 2020 statement: an updated guideline for reporting systematic reviews BMJ 2021; 372:n71 doi:10.1136/bmj.n71”

AUTHORS: Thanks for your correction. Authors have corrected this figure and reference.

REVIEWER: Figure 5: Forest plot comparative study of the effects of Cit supplementation on aerobic sports performance (≤VO2 max.).
This figure will be in in the section 3.3A Meta-analysis of Citrulline on Aerobic Performance (≤VO2 max.) of Results.

REVIEWER: Figure 6:Forest plot comparative study of the effects of Cit supplementation on the RPE This figure will be in in the section 3.3B Meta-analysis of Citrulline on the RPE of Results

REVIEWER: Figure 7:Forest plot comparative study of the effects of Cit supplementation on VO2 kinetics. This figure will be in in the section 3.3C Meta-analysis of Citrulline on VO2 kinetics of Results

REVIEWER: Figure 8:Forest plot comparative study of the effects of Cit supplementation on lactate. This figure will be in in the section 3.3D Meta-analysis of Citrulline on Lactate of Results.

TABLE

AUTHORS: Thank you for your suggestions about structure organization. As we mentioned at the beginning of this letter, we would like to be able to maintain the current structure. If this is not possible, please do not hesitate to let us know so that we can change it.

Table 1 “Design, characteristics, and gender of subjects. Type, dose, and timing of supple-mentation of the studies included in the systematic review and meta-analysis” must be moved in the Supplementary Materials.

Table 1.

effect
Create a new Table.
The first four columns of table 2, 3, 4 and 5 are the same.
Crate a new summary table, named table 1 with the first 4 columns of table 2, 3, 4 and 5, the different variables analysed and main conclusion reported in the tables 2-5.
Check population of reference number 43. There are different number in the tables 2-5.
This table will be in the section 3.2. Study Characteristics and Quality Assessment of Results.

Create a new Table 2
For each outcome must be reported
number of studies included, combined pooled SMD (Value 95% CI and p), Test of heterogeneity (Q, I2% and p) and Publication bias (Egger test)

The table 2 will be shows the results of the meta-analysis that assessed the impact of Cit supplementation on any benefits for aerobic exercise performance and specific outcomes considered (aerobic performance, on RPE, on VO2 kinetics and on lactate).
The aerobic exercise performance is the combination of aerobic performance, on RPE, on VO2 kinetics and on lactate.

Considering the characteristics of the studies,
Insert results of stratified analysis of the single outcome (aerobic exercise performance , aerobic performance, on RPE, on VO2 kinetics and on lactate) estimates according to the
- Characteristics of the subjects (Recreational, Well trained, Active
condition); Gender of subjects; Type of Citrulline supplementation (L-Citrulline, Watermelon Juice + L-Citrulline, Citrulline-Malate) and Intake time (Acute or Chronic)

Some items need to be addressed prior to publication of the study. The results are not clearly presented and some figures presented in the methods are results. I recommend the publication of present research after major revisions and done more readable the paper.
